# Characteristics of Older Adults with Alzheimer’s Disease Who Were Hospitalized during the COVID-19 Pandemic: A Secondary Data Analysis

**DOI:** 10.3390/ijerph21060703

**Published:** 2024-05-30

**Authors:** Dingyue Wang, Cristina C. Hendrix, Youran Lee, Christian Noval, Nancy Crego

**Affiliations:** 1School of Nursing, Duke University, Durham, NC 27710, USA; cristina.hendrix@duke.edu (C.C.H.); youran.lee@duke.edu (Y.L.); christian.noval@duke.edu (C.N.); nancy.crego@duke.edu (N.C.); 2GRECC Durham Veterans Affairs Medical Center, Durham, NC 27705, USA

**Keywords:** healthcare utilization, Alzheimer’s disease, COVID-19

## Abstract

We aim to investigate the relationships between the population characteristics of patients with Alzheimer’s Disease (AD) and their Healthcare Utilization (HU) during the COVID-19 pandemic. Electronic health records (EHRs) were utilized. The study sample comprised those with ICD-10 codes G30.0, G30.1, G30.8, and G30.9 between 1 January 2020 and 31 December 2021. Pearson’s correlation and multiple regression were used. The analysis utilized 1537 patient records with an average age of 82.20 years (SD = 7.71); 62.3% were female. Patients had an average of 1.64 hospitalizations (SD = 1.18) with an average length of stay (ALOS) of 7.45 days (SD = 9.13). Discharge dispositions were primarily home (55.1%) and nursing facilities (32.4%). Among patients with multiple hospitalizations, a negative correlation was observed between age and both ALOS (r = −0.1264, *p* = 0.0030) and number of hospitalizations (r = −0.1499, *p* = 0.0004). Predictors of longer ALOS included male gender (*p* = 0.0227), divorced or widowed (*p* = 0.0056), and the use of Medicare Advantage and other private insurance (*p* = 0.0178). Male gender (*p* = 0.0050) and Black race (*p* = 0.0069) were associated with a higher hospitalization frequency. We recommend future studies including the co-morbidities of AD patients, larger samples, and longitudinal data.

## 1. Introduction

Aging in place is characterized by individuals living safely, independently, and comfortably in their desired location, regardless of age, income, or ability level [1]. Approximately 80% of older adults express a preference for aging in their own homes [2]. In the older adult population, Alzheimer’s disease (AD) is a major threat to aging in place at home. The biggest risk factor for AD is old age, and 6.7 million Americans older than 65 years are currently diagnosed with AD [3,4,5]. An estimated 13.8 million people are expected to be diagnosed with AD by 2060, unless a significant breakthrough prevents, slows, or cures AD [5,6]. The morbidity, disability, and disease burden associated with AD increase the risk of hospitalizations and institutionalizations [7]. Compared to older adults without AD, those with AD have an OR = 1.65 (CI = 1.292–2.11) for hospitalization [8,9]. Approximately 20% of patients living with dementia are institutionalized within 1 year following diagnosis; this proportion increases to 50% after 5 years and approaches 90% after 8 years [10]. Worsening cognitive and functional status and behavioral symptoms such as depression or hallucinations require frequent healthcare visits before institutionalization; however, access to care was significantly impacted by the COVID-19 pandemic [11,12].

The COVID-19 pandemic has profoundly affected the US healthcare system [13,14]. The COVID-19 pandemic-associated travel restrictions resulted in dramatically fewer healthcare visits by many individuals in need [12]. Additionally, healthcare resources and attention understandably shifted towards addressing the needs of the pandemic [12,14]. Regular contact with healthcare providers is particularly crucial for individuals with AD in promoting their home-based living [12,15]. A systematic review conducted by Moynihan et al. revealed that the delivery of community-based healthcare services was reduced by about a third during the COVID-19 pandemic [16]. When healthcare resources are diverted to respond to a crisis like a pandemic, understanding the characteristics of older adults with AD who are most at risk for hospitalizations can help decision-makers to prioritize healthcare services more effectively [12,16]. The aim of this study was to examine the characteristics of older patients (65 and above) with AD who experienced at least one hospitalization during the COVID-19 pandemic. The study involved a secondary data analysis, using routinely collected in-patient electronic health record data.

## 2. Materials and Methods

We obtained a sample of 1602 people admitted to a tertiary academic health system in North Carolina with an AD diagnosis between 1 January 2020 and 31 December 2021—a period that included the COVID-19 state of emergency. This observation period spans the time in which COVID-19 was identified as a threat, the emergency declaration in the United States (US) and associated shutdowns, the flattening of the pandemic curve, virus mutations, the development and distribution of the vaccine, and the reduced COVID-19 quarantine recommendations [17,18,19]. We examined the relationships between sample characteristics and hospitalization. Sample characteristics included age, sex, race/ethnicity, insurance type, and marital status. In addition to the number of hospitalizations experienced by individuals, we also included the average length of stay (ALOS) of each hospital admission.

### 2.1. Design, Data Sources, and Study Sample

Our secondary data were obtained from DEDUCE (XX Enterprise Data Unified Content Explorer), an online query system based on data collected in the Decision Support Repository (DSR) from operational systems serving three hospitals. The applied inclusion criteria were as follows: (1) age 65 and above; and (2) AD diagnosis based on ICD10 codes (G30.0 AD with early onset, G30.1 AD with late onset, G30.8 Other AD, G30.9 unspecified AD). We excluded other types of dementia, such as vascular dementia and Lewy body dementia. This study was approved by the University Institutional Review Board.

### 2.2. Statistical Analysis

Data analysis was conducted using the Statistical Analysis System (SAS) 9.4 software [20]. We summarized the sample characteristics using descriptive statistics, and Pearson’s correlation was used to identify the relationship between age and hospitalization. For this analysis, we created a “multiple hospitalization” variable, in order to distinguish between those with more than one hospitalization and those with only one. Patients with multiple hospitalizations were categorized as “Yes” in the “multiple hospitalization” variable, while those with a single hospitalization were categorized as “No”. Independent two-sample *t*-test was conducted to determine differences in sample characteristics between those with single versus multiple (>1) admission from January 2020 to December 2021. Multiple regression was conducted to examine relationships between sample characteristics and hospitalization. The two-sided significance level was set at *p* < 0.05 in all tests.

## 3. Results

### 3.1. Sample Characteristics

Of the 1602 hospitalized adults with dementia, 1537 (95.94%) were 65 years old and above and met the eligibility criteria (Table 1). The average age was 82.20 years (SD = 7.71), and the majority were female (62.3%). Nearly two-thirds (64.1%) were White, followed by Black patients (31.2%). Most were either widowed (38.5%) or married (38.3%). The majority of patients received Medicare (62.4%), with an additional 34.6% covered by Medicare Advantage.

The mean number of hospitalizations among our sample was 1.64 (SD = 1.18). Approximately 64.2% of the sample (*n* = 987) had one hospitalization, 21.4% (*n* = 329) had two hospitalizations, and 14.4% (*n* = 221) had three or more hospitalizations. Using only the first hospitalization discharge disposition in the record, 55.1% (*n*= 847) were discharged to home and 32.4% (*n* = 498) were discharged to nursing facilities. The ALOS of all hospitalizations was 7.45 days (SD = 9.13); see Table 1.

### 3.2. Relationships between Sample Characteristics and Hospitalizations

We used Pearson’s correlation to identify the relationship between age and hospitalizations. Of those with only one hospitalization, there was no significant correlation between age and ALOS (*p* = 0.1139). Of those with multiple admissions, there was a weak negative correlation between age and ALOS (r = −0.1264, *p* = 0.0030) and the number of hospitalizations (r = −0.1499, *p* = 0.0004). The characteristics (sex, race/ethnicity, insurance type, and marital status) of those who had multiple admissions between January 2020 and December 2021 did not differ significantly from those with only one admission. 

To identify sample characteristics associated with hospitalizations, we conducted multiple linear regression analysis (for ALOS and number of hospitalizations; Table 2) and logistic regression (for Multiple Admissions (Yes/No); Table 3). Sex, race, ethnicity, marital status, insurance type, and age were significant predictors of ALOS [F (7, 1450) = 3.66, *p* = 0.0006]. Females had shorter ALOS than males (β = −1.21, *p* = 0.0227). Patients who were separated had longer ALOS (β = 2.32, *p* = 0.0056) than patients who were married. Patients with Medicare Advantage and other private insurance had longer ALOS than patients who were on traditional Medicare alone (β = 1.19, *p* = 0.0178). Sex, race, ethnicity, marital status, insurance type, and age were also significant predictors of hospitalization [F (7, 1450) = 2.97, *p* = 0.0042]. Females had fewer hospitalizations than males (β = −0.19, *p* = 0.0050). Black patients had more hospitalizations than White patients (β = 0.19, *p* = 0.0069; Table 2). There were no significant predictors of multiple hospitalizations (Table 3).

## 4. Discussion

This study explored the population characteristics of older adult patients with AD with at least one hospitalization between 1 January 2020 and 31 December 2021. The findings of this study may be helpful in guiding decision making by healthcare leaders on resource allocation and prioritization for patients with AD to promote home-based living. 

Our sample had more White, non-Hispanic females with AD. This aligns with existing findings on the development and progression of AD in females [21]. Female hormonal changes affect neural, immune, and stress interactions; consequently, the risk of AD-related morbidity rises as women age [21,22,23]. Although females had a higher prevalence of AD than males, they had fewer hospitalizations and shorter ALOS compared to males. This is consistent with a prior study reporting that male patients living with dementia had a higher frequency of hospitalizations and a longer ALOS than females [8]. 

As for discharge disposition, we observed that a higher percentage (55.1%) of AD patients were discharged to home after hospitalization, compared to what has been reported in the literature; for example, 40% was reported by Bercovitz et al. [24], and 37.6% was reported by Davis-Ajami et al. [25]. This higher home discharge disposition may have been temporarily influenced by the decreased availability of beds in long-term care facilities during the COVID-19 pandemic. During the pandemic, long-term care facilities reported shortages of personal protective equipment, staff illness, and space constraints due to isolation precautions, which reduced their capacity to accept residents [26,27,28]. 

In contrast to prior reported findings that old age contributed to a greater use of healthcare services [8,29], we found a weak negative correlation between patients’ age and ALOS (r = −0.1264, *p* = 0.0030) and the number of hospitalizations (r = −0.1499, *p* = 0.0004). The relationship between age and hospital admission in AD patients requires further examination. Older adults are more likely to be hospitalized, accounting for more than 40% of hospitalizations, although they represent only a small proportion (13%) of the US population [30]. COVID-19 hospitalization rates were highest in 2020–2021 in adults above 65 years old, with increasing age being associated with higher rates of hospitalization, peaking at 298.1/100,000 in December 2020 [18,19]. COVID-19 disproportionately affected older adults, causing severe illness and high hospitalization [31,32]. A retrospective longitudinal study of US nursing home residents above 65 years old found that 21.3% who were diagnosed with COVID-19 were hospitalized, with a higher risk of hospitalization associated with increasing body mass index, male sex, increasing cognitive impairment, and Black, Hispanic, or Asian race/ethnicity [31]. However, hospitalization risk was inconsistently associated with increasing age among US nursing home residents diagnosed with COVID-19 [31]. We did not observe age as a factor associated with hospitalizations for older adults with AD. One explanation is the interplay between the diagnosis of AD and the decision-making process of patients and family on whether to seek a hospital admission for acute problems [33,34]. AD is a progressive disease, and many of those who suffer from late-stage dementia are older adults [3]. The increasing awareness of using palliative and hospice care for incurable and late-stage illnesses in the US may help to explain our study’s results [8,29,33,34,35]. However, during the COVID-19 pandemic, the allocation of acute care resources was an additional factor that may have influenced our results [12,15,16]. For example, shortages of healthcare providers, a lack of supplies, and space constraints may result in fewer healthcare resources being allocated to older patients [36]. The use of age as a criterion in determining resource allocation disproportionately disadvantages older adults, as some strategies adopted during the COVID-19 pandemic have explicitly mentioned advanced age as a categorical exclusion when prioritization decisions are imperative [36,37]. Age alone does not capture the wide range of differences in functional ability and physiological reserve that influence morbidity and mortality in patients with AD [38]. Qualitative studies at the institutional level, such as focus groups and in-depth interviews with healthcare professionals within specific hospitals, are needed to understand how the resource allocation process may have influenced hospitalization patterns among older adults and different racial/ethnic groups during the pandemic. 

We observed that Black patients with AD were more likely to be hospitalized during the COVID-19 pandemic. This finding is consistent with prior reports that race was a risk factor in acute care utilization [31,39]. For example, compared to White patients, Black patients with AD had increased odds of Emergency Department (ED) visits and hospitalizations [40]. Pervasive health disparities in Black individuals, such as higher pre-existing rates of metabolic burden (obesity, diabetes, and hypertension) compared to non-White patients, may contribute to their increased hospitalization rates [41]. This finding is similar to those of prior studies in community residents that also noted higher hospitalization rates for Black individuals but similar mortality [42,43]. However, Black individuals hospitalized with COVID-19 were younger (average age of 60), compared to White patients (average age of 71), and 15.6% required admission to intensive care, compared to 13.9% of non-Black patients [44,45]. 

Living at home for patients with AD may be promoted if financial resources, support services, and assistance are in place from family and other support systems [46,47]. Along the same lines, those who lack social or family support (divorced or widowed) had higher odds of admission in our study. Surprisingly, we found that patients who were on Medicare Advantage and other private insurance had longer ALOS than patients who were on traditional Medicare plans. Diverse findings related to insurance groups exist in the literature, as some studies have suggested that there are no differences in ALOS between those with publicly funded insurance and those with private insurance, while others suggested a longer ALOS in publicly insured patients [48]. Individual-level as well as hospital-level characteristics can impact ALOS, to varying degrees [49]. Further investigation is needed to clarify how insurance types are associated with the length of acute care services such as hospitalization.

## 5. Conclusions

In summary, our study revealed that older adults with AD who are Black, male, and separated (widowed or divorced) had a higher risk of hospitalization during the COVID-19 pandemic compared to non-Black, female, and married older adults in the sample. These characteristics may be considered when allocating healthcare resources in times of health crises such as pandemics, when hospitalization may increase the risk of exposure to infection, especially in those with cognitive impairments such as AD [31]. If the goal is to prevent or reduce hospitalizations, more resources need to be devoted to patients with these characteristics; for example, home health nurses may need to check these patients more frequently and with shorter intervals between visits. 

This study has several strengths. It used routinely collected hospital electronic health record data to examine the associations between the characteristics of AD patients and hospitalization. The data utilized in this study were obtained from a tertiary academic hospital system across three different areas around the Raleigh–Durham–Chapel Hill area in North Carolina, providing a diverse sample population. There are also a few limitations worth noting. The nature of secondary data analysis confined us to available hospital data only. There is a body of literature that supports a strong association among AD, hospitalization, and multi-morbidity [50,51]. Unfortunately, information on co-morbidities was not included in our data, and clinical dementia ratings could not reliably be obtained from these data. We recommend that future studies include co-morbidities among AD patients and larger longitudinal samples.

## Figures and Tables

**Table 1 ijerph-21-00703-t001:** Baseline characteristics of hospitalized older adults diagnosed with Alzheimer’s disease (Total *n* = 1537).

Variables	Categories	Number (%)
Sex	Male	580 (37.7%)
	Female	957 (62.3%)
Mean Age (SD) (years)	82.2 (7.7)	
Race	White	985 (64.1%)
	Black	479 (31.2%)
	Asian and others	52 (3.3%)
	Not reported/Declined	22 (1.4%)
Ethnicity	Non-Hispanic/Latino	1474 (95.9%)
	Hispanic/Latino	34 (2.2%)
	Not Reported/Declined	29 (1.9%)
Marital status	Married	589 (38.3%)
	Widowed	591 (38.5%)
	Single	151(9.8%)
	Divorced	165 (10.7%)
	Others/Unknown	41 (2.7%)
Insurance type	Medicare	959 (62.4%)
	Medicare Advantage	531 (34.6%)
	Commercial	16 (1.0%)
	Others	31 (2.0%)
Number of Hospitalizations	One	987 (64.2%)
	Two	329 (21.4%)
	Three or more	221 (14.4%)
		Mean (SD)
Hospitalization	Count	1.64 (1.18)
Length of Stay	Average days	7.45 (9.13)

**Table 2 ijerph-21-00703-t002:** Multiple regression results: predictors of average length of stay and number of inpatient admissions of hospitalized older adults diagnosed with Alzheimer’s disease (Total *n* = 1537).

	Average Length of Stay				Number of Inpatient Admissions			
Variables	Estimate	SE	t	*p*	Estimate	SE	t	*p*
Intercept	7.43	0.49	15.07	<0.0001	1.72	0.06	27.24	<0.0001
Female (ref: male)	−1.21	0.53	−2.28	0.0227 ^±^	−0.19	0.07	−2.81	0.0050 ^±^
Black (ref: White)	0.44	0.54	0.81	0.4169	0.19	0.07	2.70	0.0069 ^±^
Non-White Non-Black (ref: White)	−2.11	1.68	−1.26	0.2092	0.06	0.21	0.27	0.7889
Hispanic (ref: Non_Hispanic)	−0.12	2.56	−0.05	0.9635	−0.30	0.33	−0.92	0.3563
Single * (ref: married)	−0.04	0.56	−0.07	0.9436	0.05	0.07	0.69	0.4877
Separated ** (ref: married)	2.32	0.84	2.78	0.0056 ^±^	0.08	0.11	0.72	0.4721
Other insurance (ref: medicare)	1.19	0.50	2.37	0.0178 ^±^	−0.11	0.06	−1.75	0.0802

* Single: never married; ** Separated: include divorced and widowed. ^±^ Statistically significant.

**Table 3 ijerph-21-00703-t003:** Logistic regression results: odds ratios of multiple admissions (Yes/No) of hospitalized older adults diagnosed with Alzheimer’s disease (total *n* = 1537).

	Multiple Admissions (Yes/No)			
Variables	OR	95% CI		*p*
Female (ref: male)	0.869	0.686	1.101	0.2451
Black (ref: White)	1.206	0.949	1.533	0.4699
Non-White Non-Black (ref: White)	1.062	0.499	2.261	0.9309
Hispanic (ref: Non-Hispanic)	0.327	0.072	1.492	0.1490
Single * (ref: married)	1.003	0.781	1.287	0.5922
Separated ** (ref: married)	0.879	0.603	1.281	0.4610
Other insurance (ref: Medicare)	0.880	0.702	1.102	0.2655

* Single: never married; ** Separated: include divorced and widowed.

## Data Availability

Our secondary data were obtained from DEDUCE (Duke Enterprise Data Unified Content Explorer), an online query system based on data collected in the Decision Support Repository (DSR) from operational systems serving three hospitals. Link: https://sites.duke.edu/giminternal/research/pace-deduce-maestro-care-slicer-dicer-tool/ accessed on 17 November 2022.

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
