# Peer review of "Characteristics of Older Adults with Alzheimer’s Disease Who Were Hospitalized during the COVID-19 Pandemic: A Secondary Data Analysis"

_ijerph, 2024, doi:10.3390/ijerph21060703_

Round 1
Reviewer 1 Report
Comments and Suggestions for Authors
Report Reviewer IJERPH (ISSN 1660-4601)
Manuscript ID: ijerph-3001913
Title: Characteristics of Older Adults with Alzheimer’s Disease Who Were Hospitalized During the COVID-19 Pandemic: A Secondary Data Analysis
Comments and Suggestions for Authors
The authors aimed to understand relationships between population characteristics of patients with Alzheimer’s disease (AD) and their Health Utilization (HU) during the COVID-pandemic. The study found that older adults with AD who are Black, male, and separated (widowed or divorced) had more risks for hospitalization during the COVID-19 pandemic compared to non-Black, female, and married older adults in the sample.
The research is comprehensive and detailed, has good figures and I believe would be of interest to many readers. The topic is very popular and of interest to the field. Some suggestions for the improvement are listed below:
1. The manuscript needs significant grammatical editing throughout. Ungrammatical English phrases are widespread in the text. Although they do not necessarily prevent the reader from understanding the text, the text would be improved if it were reviewed and revised for its grammar. Some typos and grammatical errors were observed, resolve them.
2. Some relevant references can be added to the introduction because they serve the importance of this topic; http://dx.doi.org/10.22159/ijap.2020v12i5.38439
3.The abstract is too long. It should be shortened.
4. The doi numbers of some the references are not mentioned like ‘[43-48], Provide the same.
-Words in article titles are written in upper or lower case letters. This needs to be improved.
.
Comments on the Quality of English Language
The manuscript needs significant grammatical editing throughout. Ungrammatical English phrases are widespread in the text. Although they do not necessarily prevent the reader from understanding the text, the text would be improved if it were reviewed and revised for its grammar. Some typos and grammatical errors were observed, resolve them.
Reviewer 2 Report
Comments and Suggestions for Authors
a. Abstract:
Background:
- ”We aim to understand relationships between population characteristics of patients with Alzheimer’s Disease (AD) and their Health Utilization (HU) during the COVID-19 pandemic" - This sentence could be revised for clarity. Consider: "We aim to investigate the relationship between population characteristics of Alzheimer’s Disease (AD) patients and their healthcare utilization (HU) during the COVID-19 pandemic."
Methods:
- ”EHR from a North Carolina tertiary institution was utilized." - Specify "electronic health records (EHR)" for clarity.
- "between 1/1/2020-12/31/2021" - Use "from" instead of "between" for clarity.
Results:
- ”with an average age of 82.20 years (SD=7.71)" - Add a space before "years" for consistency.
- "Discharge dispositions were primarily home (55.1%) and nursing facilities (32.4%)." - Consider adding "to" before "nursing facilities" for clarity.
Conclusions:
- ”requires further examination" - Consider specifying what aspect requires further examination for clarity.
- "We recommend future studies include co-morbidities of AD patients" - Add "that" after "studies" for grammatical correctness.
b. Introduction:
- ”Almost 80% of older adults identified their home as their desired place to age" - Consider rephrasing for clarity. For example, "Approximately 80% of older adults express a preference for aging in their own homes."
- "The biggest risk factor for AD is old age" - Specify "Alzheimer’s disease (AD)" for clarity.
- "are diagnosed with AD today" - Consider specifying the timeframe for clarity, such as "are currently diagnosed with AD."
- "13.8 million people are expected to be diagnosed with AD in 2060" - Specify "Alzheimer’s disease (AD)" for clarity.
- "Compared to older adults without AD" - Specify "Alzheimer’s disease (AD)" for clarity.
- "this proportion increases to 50% after 5 years and approaches 90% after 8 years" - Clarify what "this proportion" refers to.
- "but access to care was significantly impacted by the COVD-19 pandemic" - Correct "COVD-19" to "COVID-19" for accuracy.
- "Moynihan and colleagues" - Provide the full reference for clarity.
- "For people with AD, this was especially impactful as regular contacts with health care providers are critical elements in promoting home-based living." - Consider rephrasing for clarity, such as "Regular contact with healthcare providers is particularly crucial for individuals with AD, especially in promoting home-based living."
c. Results:
- ”Of the 1602 hospitalized adults with dementia, 1537 (95.94%) were 65 years and above" - Consider specifying "years old" for consistency.
- "Almost two-thirds (64.1%) were White followed by Black (31.2%)" - Consider specifying "White individuals" for clarity.
- "Discharged to nursing care facilities" - Consider rephrasing for clarity, such as "discharged to nursing facilities."
- "A multiple admission variable was created for this analysis" - Specify what "multiple admission variable" refers to for clarity.
- "There were no significant predictors of multiple hospitalizations" - Specify what "multiple hospitalizations" refer to for clarity.
d. Discussion:
- ”This is consistent with reported findings that AD development and progression are related to changes in female hormonal, neural, immune, and stress interactions which increases the risk of AD-related morbidity as women age" - Consider breaking this sentence into smaller sentences for clarity.
- "Although females have higher AD prevalence than males, they had fewer hospitalizations and shorter ALOS compared to males" - Specify "Alzheimer’s disease (AD)" for clarity.
- "In contrast to prior reported findings" - Specify what prior findings are being referenced for clarity.
- "However Black individuals hospitalized with COVID-19 were younger" - Correct "However" to "However, black individuals hospitalized with COVID-19 were younger" for grammatical correctness.
- "The increasing awareness of palliative and hospice care by many in the US for incurable and late-stage illnesses" - Consider rephrasing for clarity.
- "However, during the COVID-19 pandemic, allocation of acute care resources was an additional factor that may have influenced our results" - Consider specifying how the allocation of acute care resources influenced the results.
- "The lack of robust triage and decision-making frameworks" - Specify "decision-making frameworks" for clarity.
- "Qualitative studies of organization-level resource allocation processes" - Specify "organization-level" for clarity.
e. Conclusions:
- ”may benefit decision-makers prioritizing health care services" - Consider rephrasing for clarity.
- "If the intent is to keep people from hospitalization" - Clarify what "the intent" refers to.
- "There are several strengths of this current study" - Consider rephrasing for clarity, such as "This study has several strengths."
- "A few limitations are worth noting" - Consider specifying what the limitations are for clarity.
Scientific Comments:
- The article effectively addresses the relationship between population characteristics of older adult patients with Alzheimer's disease (AD) and their healthcare utilization during the COVID-19 pandemic. The study design, data analysis methods, and results interpretation are appropriate for addressing the research questions.
- The introduction provides a comprehensive overview of the significance of AD, aging in place, and the impact of the COVID-19 pandemic on healthcare utilization. However, some statements could benefit from additional citations to support the claims made.
- The results section presents key findings regarding sample characteristics, hospitalizations, and predictors of healthcare utilization among older adults with AD. The statistical analyses performed are appropriate, and the results are clearly presented.
- The discussion section effectively interprets the study findings in the context of existing literature. However, some statements lack citations to support the claims made, especially regarding the impact of the COVID-19 pandemic on healthcare utilization among individuals with AD.
- The conclusions draw appropriate implications from the study findings and highlight the importance of considering population characteristics when allocating healthcare resources during health crises such as the COVID-19 pandemic. However, the recommendations could be strengthened by providing specific strategies for addressing the identified disparities in healthcare utilization.
Moderate editing of English language required
Reviewer 3 Report
Comments and Suggestions for Authors
Basic reporting
- This paper is an observation study that analyzed the demographics of patients with Alzheimer’s disease (AD) during COVID pandemic.
Experimental design
Strengths of the paper
1. The paper includes clearly describes the introduction, methods, and major findings.
2. The paper performs sound statistical analysis and reports the details enough to replicate the study.
Weaknesses
1. One minor recommendation I have is to use the terms Caucasian and African American instead of White and Black.
2. Inclusion criteria should mention the criteria used to diagnosis AD.
3. The paper should also make a comment if patients with AD also had other dementia in addition of AD (which is expected given the mean age of the participants.
Additional comments
The paper involves sound methodology and gives enough details to be replicable. I recommend acceptance after minor revisions.
Comments on the Quality of English LanguageBasic reporting
- This paper is an observation study that analyzed the demographics of patients with Alzheimer’s disease (AD) during COVID pandemic.
Experimental design
Strengths of the paper
1. The paper includes clearly describes the introduction, methods, and major findings.
2. The paper performs sound statistical analysis and reports the details enough to replicate the study.
Weaknesses
1. One minor recommendation I have is to use the terms Caucasian and African American instead of White and Black.
2. Inclusion criteria should mention the criteria used to diagnosis AD.
3. The paper should also make a comment if patients with AD also had other dementia in addition of AD (which is expected given the mean age of the participants.
Additional comments
The paper involves sound methodology and gives enough details to be replicable. I recommend acceptance after minor revisions.
